# Calculating forest species diversity with information-theory based indices using sentinel-2A sensor's of Mahavir Swami Wildlife Sanctuary

**Pavan Kumar** *, **Manmohan Dobriyal, Amey Kale, A. K. Pandey, R. S. Tomar, Elizabeth Thounaojam**

College of Horticulture and Forestry, Rani Lakshmi Bai Central Agricultural University, Jhansi, India

* pawan2607@gmail.com

**Data Availability Statement:** All relevant data are within the paper and its Supporting Information files.

## Abstract

Tropical forest serves as an important pivotal role in terrestrial biological diversity. The present study makes an attempt to identify the concentration of species among tree diversity in Mahavir Swami Wildlife Sanctuary, Bundelkhand, India. Four important ecological indicator indices namely Shannon-Weiner index (H'), Simpson's diversity (D), Margalef index (SR) and Pielou's (J) indices were make the most for species diversity measurement. The research outcomes revealed that Shannon-Weiner diversity index (H′) was found to be the best index for assessing species richness while Simpson's diversity (D) index was more suited for determining species diversity. The Shannon-Weiner index value calculated for different transects not only represent the species richness but also the species evenness in each transect. The potential application of forest diversity can be used a mechanism for forest management. The methodology will retrofit better policy implementation for maintaining the health of forest species in Mahavir Swami Wildlife Sanctuary and can be applied on other reserve forest of socio-ecological significance.

## 1. Introduction

Forests are at constant risk due to its transformation for agriculture and pastures and its exposure to forest fire, pest and disease outbreak. All these threats become sources of $CO_2$. However, it is not always the case. Some areas under forest are being converted to plantations and increasingly becoming carbon sinks [1–3]. Emissions of $CO_2$ as a result of transformation of forest into agriculture were higher than the burning of fossil fuels prior to this century [4,5]. Forests are significant as these reduce concentration of carbon dioxide in the atmosphere and help in mitigating climate change. Therefore, management of forest resources can help in conserving biodiversity, water and soil within forest ecosystem. Forests capture carbon from the atmosphere through the process of photosynthesis [6]. These convert photosynthate into biomass through the processes of respiration and decomposition and emit carbon back into the atmosphere. Forests can play an important part in preventing increase of greenhouse gases in

**Funding:** The author(s) received no specific funding for this work.

**Competing interests:** The authors have declared that no competing interests exist.

the atmosphere. Management and monitoring of forest is therefore, is important for reducing the rate of increased $CO_2$ in the atmosphere. The main aim of forest management is to increase vegetation so that the amount of carbon may increase. The carbon pool in vegetation can be accomplished through protecting degraded forest to get the maximum value of carbon sequestration [7]. Agro forestry programme by bringing non-forested areas under forest is indicative of accomplishing such task. Forest transformation is the major source of anthropogenic induced carbon dioxide emissions and is responsible for generating nearly 10 to 25% $CO_2$ globally [8].

Tropical forests are the important source for carbon pool and contain about 40% of the terrestrial carbon storage. These forest account for 1/3rd of global net primary productivity. Understanding of species richness, ecological and structural traits are indelible for conserving the forest diversity [9]. There are various indices for assessing the diversity and richness of species namely Shannon-Wiener index, Margalef index, Simpson index, etc. Shannon diversity index was mainly used for identifying richness and diversity of trees species [10]. In 1958 Margalef has markedly popularized concept of species diversity among the scientific community [11,12]. Assessment of species evenness, diversity and richness has found instructive for future researches in various forest ecosystems at spatial scales.

Species diversity may be is defined as variability of living organisms in ecosystems. It may also be taken as diversity between species and within species in a particular ecosystem [13,14]). It is essentially considered as significant component of ecosystem as it helps in hydrological and climatic regulation [15]. Species diversity is on decline globally in spite of enactment of Convention on Biological Diversity (CBD). Most of the biodiversity hotspots are found in the tropics [16]. These hotspots are characterized by having a large number of endemic and endangered species. However, information on vegetation diversity and its significance in the ecosystem of tropical forests is scant [17]. Assessment of biodiversity in tropical forests is challenging task mainly due to climatic conditions, inaccessibility and complex diversity. Monitoring changes in biodiversity, however, is essential for effective conservation planning [18]. Diversity in species and their stand structure assume greater significance because tree species are helpful in providing basic needs and habitat for other species. It has been established that distribution of species, their structure and response to ecosystem are essentially key factors for ecological study. Further, an understanding of the diversity and stand structure of species is prerequisite for regulating climate change. Hence, monitoring of forest is an essential to assess sustainability in species diversity.

Remote sensing data has proved productive in examining the heterogeneity of the resources and their spectral signatures [19]. Species are one of the significant elements of biodiversity. The network structure of species supplements information on the species richness. High species heterogeneity refers to the higher number of species [20]. Analyzing spatial elements of biodiversity and causative environmental factors are now been gaining interest among ecologists [21]. The studies with the information derived from spectral data variability can be seamlessly beneficial for assessing the diversity based on productivity and habitat structure of species [22,23]. The potential of earth observation (EO) as an effective tool for exploring and monitoring patterns of forest species diversity had been largely ignored until the early 2000s, as previous studies have thoroughly reviewed [24–27].

Relationship between diversity and different vegetation indices are widely used for assessing the species richness. Estimation of the DBH with the help of species diversity are correlated with both species canopy cover and tree richness [28] in their study correlated vegetation with DBH [29] measured biomass production using integrated normalized difference vegetation index (INDVI). No nascent effort till date has been acknowledged by forest ecologists to measure the biodiversity richness in any part of forest by using NDVI and DBH. NDVI has been

associated to DBH at wide-ranging longitudinal gauges [30]. The relationship between DBH and NDVI as well as biodiversity and species richness indicates strong relationship among species richness and NDVI.Structure of species and complexity of habitat is a relative parameter to species richness which can be evaluated different spectral indices. Spectral variations are closely associated to heterogeneity of resource distribution which can be analyzed through differences of spectral signatures of remote sensing data [22]. Satellite data helps to identify the occurrence, manifestation and spatial distribution of species. These data can be useful in calculation of net primary productivity (NPP) using NDVI [31]. Occurrence of species and their richness at various scales can be modelled by using NDVI. High heterogeneity of tree species refer to high existence of tree species [12,20,32,33].

Various diversity indices are used for measuring attributes of community structure [34,35]. Information on rarity and commonness of trees species can be derived through diversity indices. These indices can also help in comparing different habitat types and individual habitat. The commonly used diversity indices include Shannon wiener diversity index [36–38], Whittaker index of species evenness [39], Margalef index of species richness [40–42], Simpson's diversity index [38,43], Alfa diversity exponential Shannon wiener index, Bray-Curtis index [44] and Alfa diversity exponential Shannon wiener index [45].The present study makes an attempt to calculate species diversity indices and structural forms of the tropical forest.

## 2. Material and methodolohy

### 2.1 Study area

In the Indian state of Uttar Pradesh, Mahavir Swami Wildlife Sanctuary (MSWS) is situated in the Lalitpur. Its latitudes and longitudes extent is 24˚29' N to 24˚32' N and 78˚14' E to 78˚17' E. situated at an elevation of 300 ft above Betwa river, near the Vindyan Range. It is 125 km away from Jhansi (Fig 1). It spreads over an area of 5.4 km$^2$. The total geographical area is 13.79 h. The annual dry months may range from 3 mm to 7 mm. The summer temperature ranges from 27˚C to 45˚C and may exceed 48˚C and winter temperature ranges from 6˚C to 26˚C.The Sanctuary is lesser known for its wild animals, but an important site for the critically endangered vulture species and focused on their populations and conservation strategy. It has been an important breeding sitefor the Long-billed vultures *(Gyps indicus)* and is being monitored regularly since 2007. The high cliffs are an ideal habitat for the vultures, owls, eagles and other bird species. Different species in reptile like magar, python snake, chinkara, wolf, wild cat, hyena, wild dog, chinkara, etc.Primarily the forest comprised of Sagwan*(Tectona grandis)* and other trees species like Arjun *(Terminalia arjuna)*, Tendu *(Diospyros melanoxylon)*, Mahua *(Madhuca Indica*, *Chironji(Buchananialanzan)* are also found. There is also lower strata of shrubs and climbers in many pockets.

### 2.2 Sentinel-2A data based biodiversity extent

Multi spectral instrument (MSI) of Sentinel-2A satellite data with twelve spectral bandsand 10 meters spatial resolution was procured on April 2021 for developing NDVI indices of the sampled tree species [46]. The effectiveness of Sentinel-2A data in monitoring land use at optimal ground resolution help in monitoring land use/land cover changes due to wildfire, forest change, drought, urbanization, climate change, etc. Atmospheric and geometric corrections of satellite image were carried out using image processing software ERDAS IMAGINE (v. 2014). False colour composite (FCC) was developed for interpretation of different land use/cover categories using element image interpretation. Delineated tree species on satellite image were verified with geo-tagged surface species through ground truth.

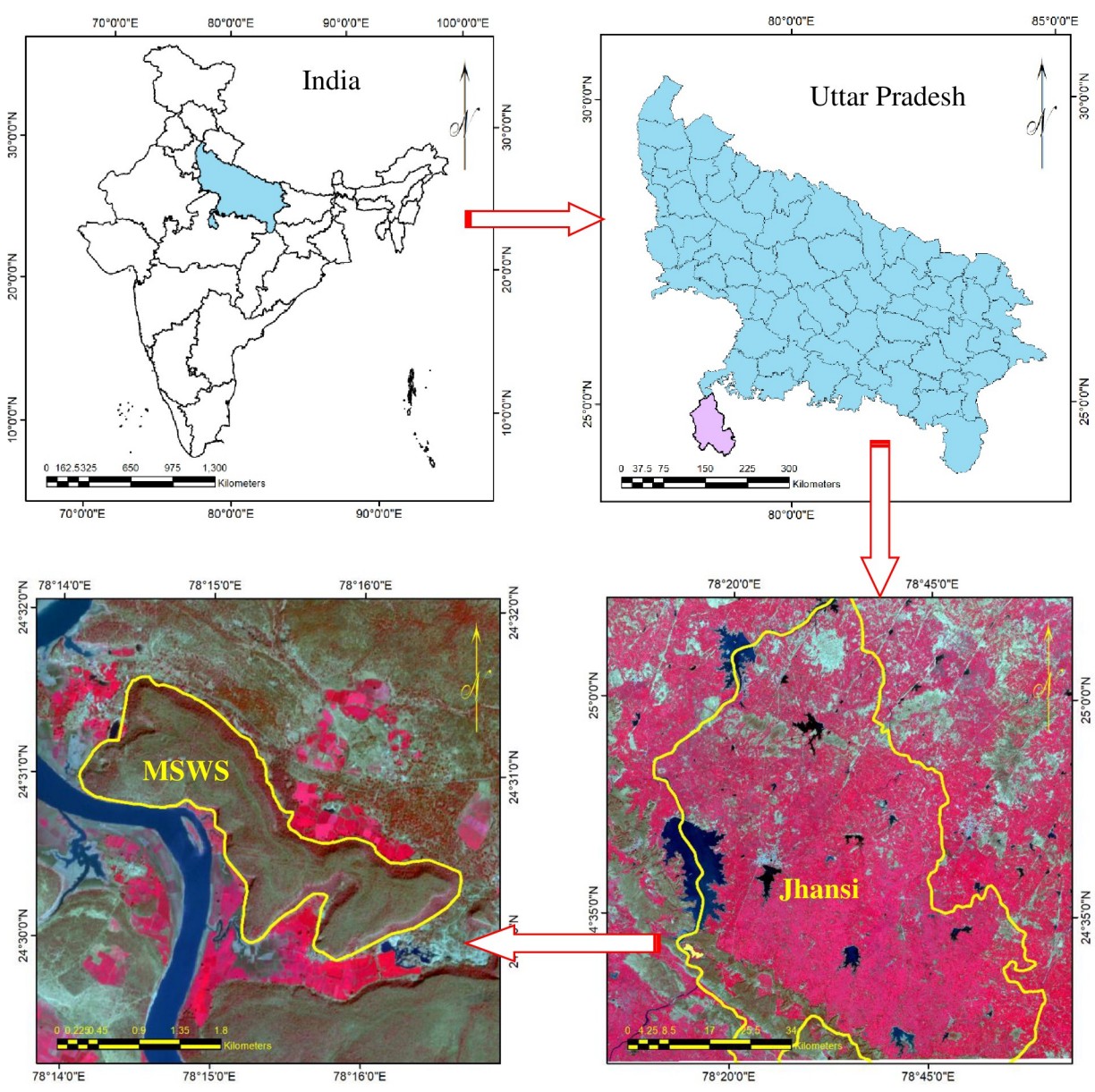

**Fig 1. Study area. (Source:** https://www.usgs.gov/centers/eros).

To find out biodiversity extent based on Sentinel-2A sensors's, we coverd two separate groups of studies including biodiversity extent and forst density (Fig 2C and 2D). The first is forest density mapping in a perspective of demographic zone (Fig 2A), by linking with census data (Fig 2B). The second one is generated biodiversity extent (Fig 2D). Biodiversity extent indicates the boundary that separates forest areas from surrounding biodiversity extent based on Sentinel-2A sensors's. We included forest density mapping in this study. Both temporal and spatial dimensions of sensors's data have been widely explored for forestcovermapping.

### 2.3 Selection of samples and transect layout design

Dimension of samples (Fig 3) and unpredictability in species affect the accuracy of their estimation. Population is divided into homogenous groups and samples are selected randomly

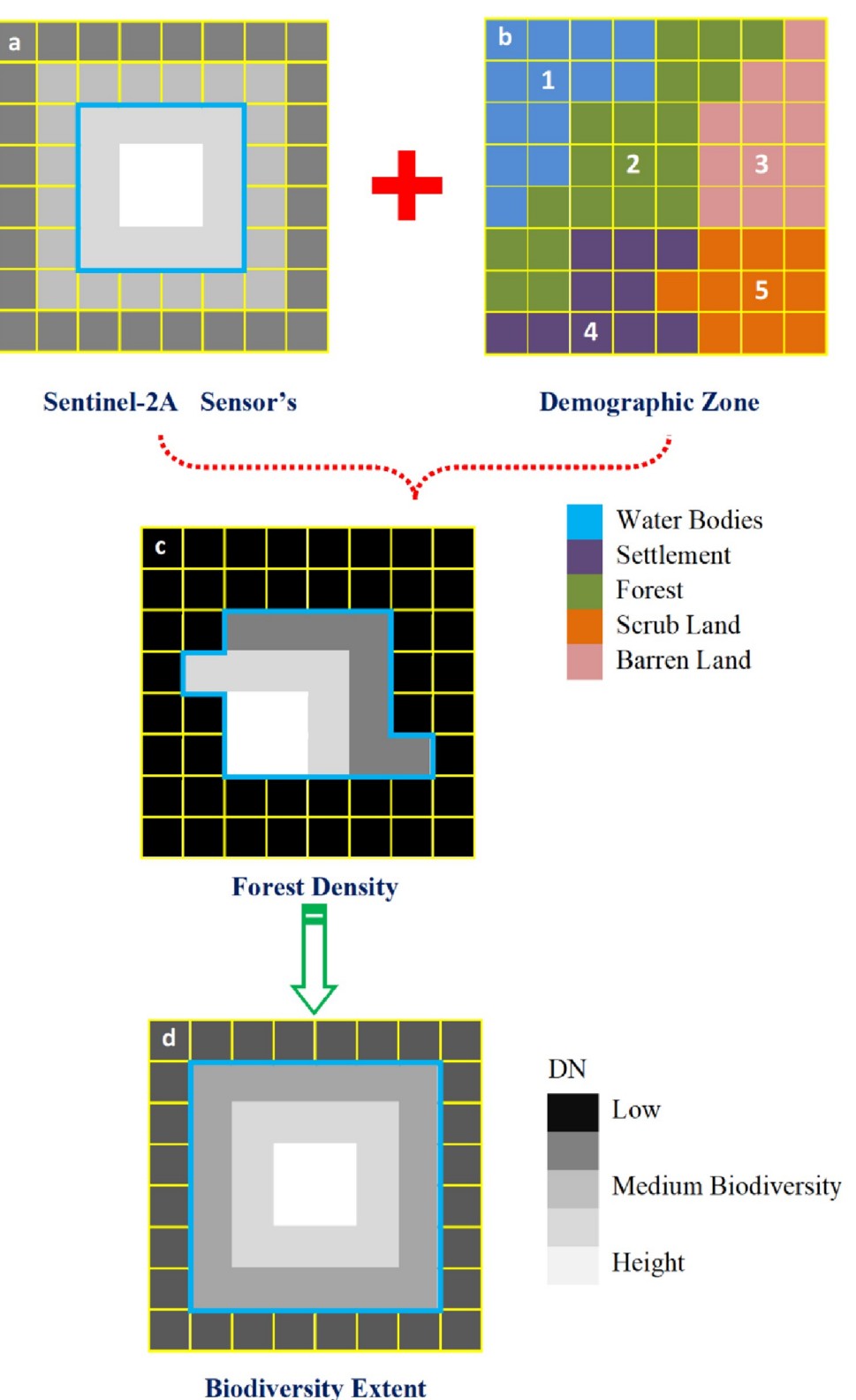

**Fig 2. Sentinel-2A based biodiversity extraction.**

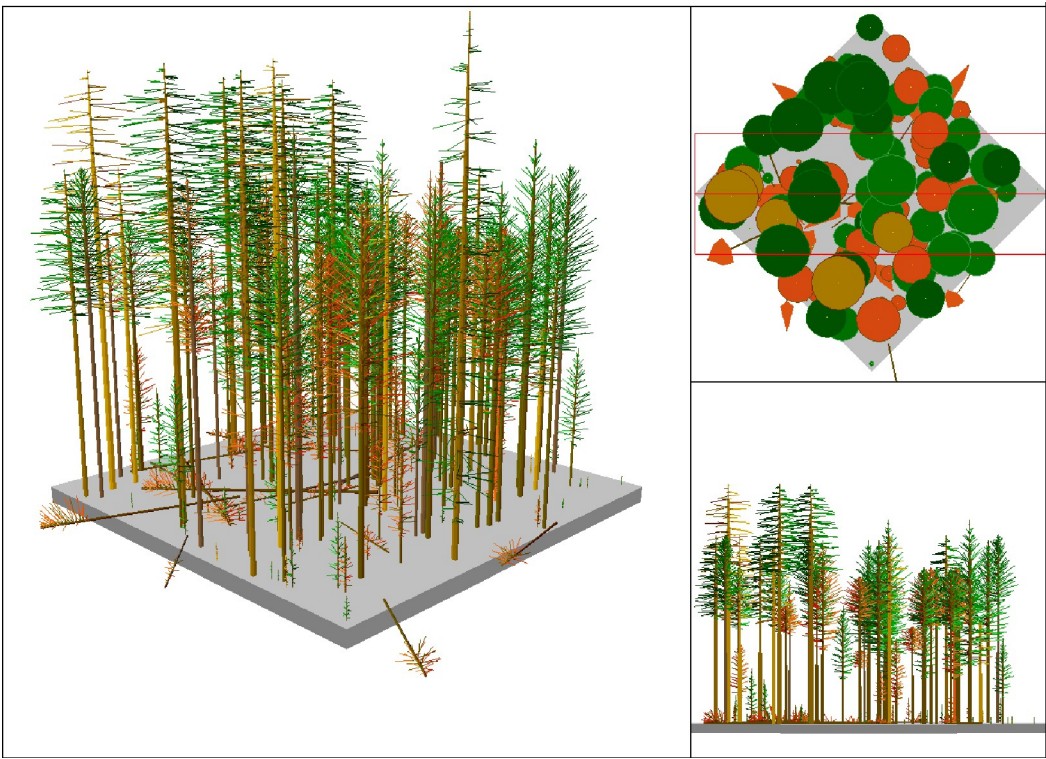

**Fig 3. Transect layout design.**

from such group independently for reducing sampling discrepancy [47]. Effectiveness of stratification can be assessed through variance analysis. Permission was obtained from Divisional Forest Officer (DFO), Kaimoor Wildlife Division, Mirzapur, Uttar Pradesh, India. Stratified random sampling technique was utilized for collecting samples from the homogeneous groups having high normalized difference vegetation index (NDVI) values. Size of the sample was determined as:

$$N = \left( CV * \frac{t}{e} \% \right)^2 \qquad (1)$$

Where:

N = Number of samples needed

t = Student's 't' at a given level of probability (2 for 95% confidence limit)

e = Desired sampling error expressed as % of mean

CV = co-efficient of variation (%)

The total required transect were determined to be 13assuming that E% is equal to 10%. Threetransectswere located in inaccessible areas. Hence, 10 transects were collected from the study area. Topographic sheet on a scale of 1:50,000 were used for preparing base map and designing transect. For this, the topographic map was divided into number of grids of 2½' × 2½'. Four sub grids of 1¼' × 1¼' size were made within each such grid of topographic map. Sufficient care is taken in laying of transects. The aspect of all the transects should be the same.

The transect should be laid in a frame of 31.62*31.62 meter on one side of the stream avoiding crossing any stream area. Uniform direction either left or right should be followed throughout the transects. All the materials should be kept outside the ransect to avoid trampling of the flora.

## 2.4 Random variable selection probability for each species

Selection of species of ten transect in the study were measured by applying rule of random variable probability selection method. The theorem of marginal distribution was applied to select the parameter for Akaike's Information Criterion (AIC) model prediction. For AIC model, if $P_x$ and $P_y$ be the probability measures induced by $X$ and $Y$, respectively, on $(R^1, B^1)$ then $P_{xy}$ is supposed to be probability measure induced by the two dimensional random variable (X,Y) on $(R^2, B^2)$. Let $B_1$ and $B_2$ be arbitrary sets (Borel) in the spaces $X$ and $Y$, respectively:

$$P_{XY}(B_1 X R^1) \qquad (2)$$

The probability that (X,Y) takes a value in $B_1$ X $R^1$ or the probability that X takes a value in $B_1$ irrespective of the value taken by Y. Similarly, we can use the formula to predict the species:

$$P_{XY}(R^1 X B_2) \qquad (3)$$

The Eq (3) represents the probability that (X, Y) takes a value in $R^1$ X $B_2$ or the probability that Y takes a value in $B_2$ irrespective of the value taken by X. Accordingly, from relation (2) and (3) we can have the final function to choose the species from all the transects as:

$$P_{XY}(B_1 X R^1) = P_X(B_1) \qquad (4)$$

$$P_{XY}(R^1 X B_2) = P_Y(B_2) \qquad (5)$$

The probabilities $P_{xy}(B_1 X R^1)$ for varying $B_1 \varepsilon B^1$ are said to be the *marginal distribution* of X relative to the joint distribution of X and Y. Clearly, this can be obtained by projecting the mass of the joint distribution on the subspace of different variables. Similarly, the probabilities $P_{xy}(R^1 X B_2)$, for varying $B_2 \varepsilon B_1$ can defined as the marginal distribution of Y. Individual species were selected from 10tarnsects using the comparative relation of Eqs (4) and (5).

## 2.5 Empirical analysis of diversity indices

The lack of sufficient sampling in any forest area has become the main challenging task for biologist to acquisition of species richness of any forest area (e.g., [48]. In date, to minimise such a problem like sampling and resulting complication using species richness indices are the most inventorying way [49,50]. According to [51], remote sensing is the tool for estimating the species richness has been widely used and recognized as one of the most promising approach. The use of remote sensing for determination of species richness minimizes the divergence of difficulties which is arising in field based data collection. The discrepancy arises on field data is minimal by using the remote sensing data which is a good proxy of biodiversity at any species level.

**2.5.1 Shannon Weiner index ($H'$)..**   Shannon's index ($H'$) was developed to findoutspecies diversity [52]):

$$H' = \sum_{i=1}^{S} p_i \ln p_i \qquad (6)$$

Where:
$H'$ = Shanon-Weiner index
$p_i$ = Proportion of individual belonging to species i
ln = Natural log

**2.5.2 Margalef index (SR).** Margalef index was utilized for estimation of species richness [53]:

$$SR = \frac{S - 1}{\ln(N)} \quad (7)$$

Where:

*SR* = Margalef index of species richness

*S* = Number of species

*N* = Total number of individuals

**2.5.3 Simpson's index (D).** Simpson's index was evaluated for determining species diversity. Plant diversity increases by species richness and their regularity:

$$D = 1 - \frac{\sum n(n - 1)}{N(N - 1)} \quad (8)$$

Where:

*n* = Total number of organisms of a particular species

*N* = Total number of organisms of all species

**2.5.4 Pielou's Index (J).** Species evenness was determining to findoutPielou index [44]:

$$J = \frac{H^{/}}{\ln(S)} \quad (9)$$

Where:

*J* = Pielou's measure of species evenness,

$H^{/}$ = Shannon-Wiener index,

*S* = Total number of species/sample

## 2.6 Visual interpretation of species

Forest ecosystem can best be described by diversity in its tree species. Tree species diversity is a significant factor for modelling wildlife habitat and assessing effective management of forest resources [49,54]. Thus, analysis of spatial distribution of species is an important aspect for forest management. Traditional forest inventories and tree species stand estimation are not sufficient for such task. Tree species information over a large area can not be obtained on the basis of only field data. Therefore, sophisticated methods are required for collecting information of spatial distribution of tree and their composition characteristics.Remote sensing sensors having various spatial and spectral resolutions provide information on forest over a larger area for analyzing finer details of forest species [12,18]). The effect of spatial resolution on pixel color and visual interpretation of the tree speciesin the image has been shown in Fig 4. In the figure, heterogenity of the image area has been shown over 100 x100 m area of MSWS at 1m resolution in the top left pixelcontext. In the middle row, the image variation from lowest resolution (1m) to highest resolution (0.4mm) has been shown. The highest resolution image interpretability has been presented at top right corner in the figure. As the resolution increases the visual identification of the tree species becomes more effective. The variation in pixel color has also been shown in pixel color row with increasing resolution.

## 3. Results and discussion

Tropical forest acts as munificence for the life forms living in the tropics, by providing habitat conditions and natural resources. Analytical scrutiny of the forest stand is provided by the

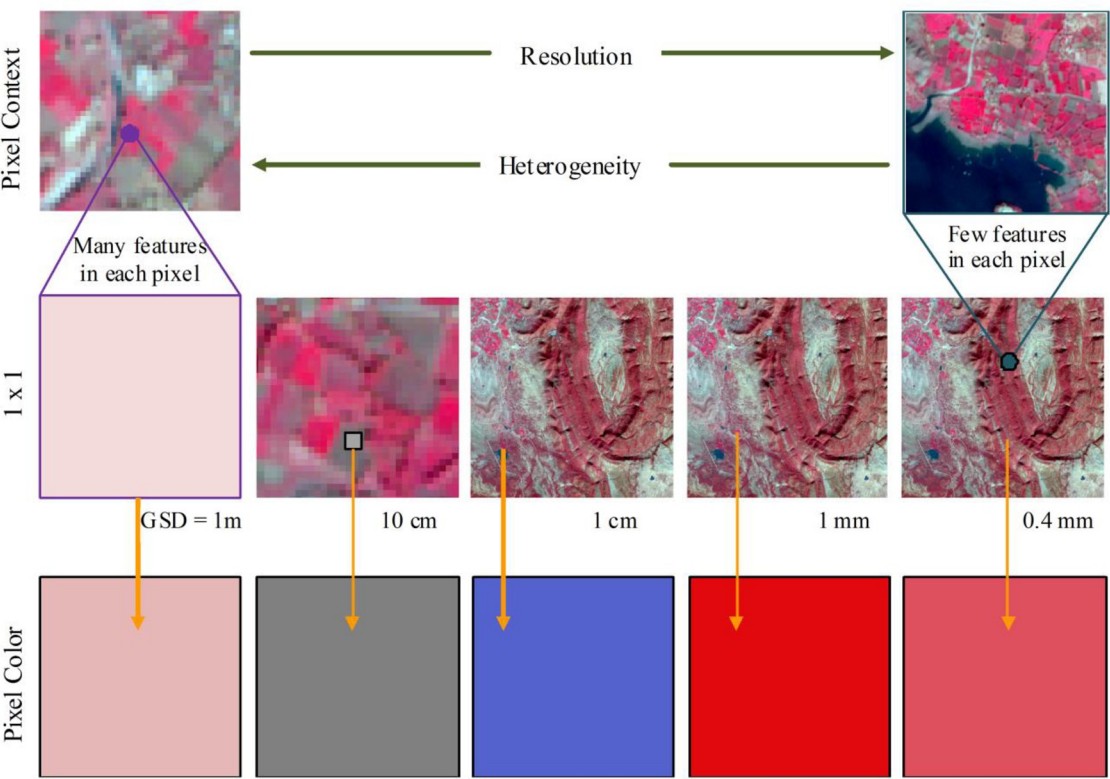

**Fig 4. Effect of spatial resolution on visual interpretation of the Image. (Source:** https://www.usgs.gov/centers/eros).

structural diversity of forest and can be sub divided into three categories; tree species diversity, tree dimension diversity and tree position diversity. Forest is the key indicator of current study. The commonly used diversity indices include Shannon Weiner index ($H'$), Simpson's diversity ($D$), Pielou's (J) and Margalef index ($SR$)index were utilized to assess tree species diversity.

## 3.1 Species diversity

Ten transects of 0.1 ha were delineated in foorest area and selected 16 tree species and 539 individuals in all transect.A total of 16 tree species belonging to 13 genera and 11 families have been recorded. In this way 539 individuals were recorded in 10 transects 0.1 ha. Other species were few in numbers and occurred as associate species forming mixed tree species. Transect-wise species record has been presented in Table 1.

**Table 1. Transect-wise species recorded in MSWS.**

| Variables | $T_1$ | $T_2$ | $T_3$ | $T_4$ | $T_5$ | $T_6$ | $T_7$ | $T_8$ | $T_9$ | $T_{10}$ |
|---|---|---|---|---|---|---|---|---|---|---|
| Total no. of species (S) | 4 | 10 | 3 | 5 | 3 | 7 | 9 | 6 | 4 | 7 |
| Total no. of Individual (N) | 11 | 54 | 31 | 47 | 32 | 72 | 83 | 65 | 54 | 90 |
| Natural log of species (ln S) | 1.38 | 2.30 | 1.09 | 1.60 | 1.09 | 1.94 | 2.19 | 1.79 | 1.38 | 1.94 |
| Natural log of induals (ln S) | 2.39 | 3.98 | 3.34 | 3.85 | 3.46 | 4.27 | 4.41 | 4.17 | 3.98 | 4.50 |

Source: Based on field survey and Google earth.

**Table 2. Diversity indices recorded in MSWS.**

| Variable | $T_1$ | $T_2$ | $T_3$ | $T_4$ | $T_5$ | $T_6$ | $T_7$ | $T_8$ | $T_9$ | $T_{10}$ |
|---|---|---|---|---|---|---|---|---|---|---|
| Margalef index of species richness (*SR*) | 1.25 | 2.25 | 0.58 | 1.03 | 0.57 | 1.40 | 1.81 | 1.19 | 0.75 | 1.33 |
| Simpson's diversity index (*D*) | 3.90 | 6.53 | 1.70 | 3.05 | 1.79 | 4.32 | 5.29 | 3.67 | 2.67 | 4.81 |
| Shannon-Weiner index (*H'*) | 1.37 | 2.06 | 0.74 | 1.32 | 0.78 | 1.66 | 1.87 | 1.49 | 1.11 | 1.69 |
| Pielou's Index (J) | 0.99 | 0.89 | 0.67 | 0.82 | 0.71 | 0.85 | 0.85 | 0.83 | 0.80 | 0.87 |

Source: Researchers'calculation.

## 3.2 Tree species indices

Attributes of community structure as ecological indicators can be utilized for deriving various diversity indices for examining tree species. Commonness of species and rarity in a community can be obtained by using different diversity indices. Different habitat types then can easily be compared with the help of these indices [55]. Species richness, diversity and evenness indices were calculated for each transect. Maximum number of species have been recorded along with their even distributionintransect $T_2$, $T_6$, and $T_{10}$ as evident from Shannon-Weiner index (H') results(Table 2). Very less number of species have been recorded in transect $T_3$, $T_5$ and $T_9$resulting into lower values for diversity indices due to consideration of both number of species recorded and their relative abundance in each transect.

**3.2.1 Margalef index.** The value of Margalef index of species richness for the transect $T_1$ to $T_{10}$varied between 0.57 and 2.25 as shown in Table 2. Higher values have been recorded in transects $T_2$ and $T_6$, *i.e.*, 2.25 and 1.40 respectively, whereas comparatively lower values were recorded for transect $T_5$ and $T_3$*i.e.*, 0.57 and 0.58.The higher valuesrecorded for Margalef index of species richness were directly related to the number of species present in each transect. Different diversity indices were calculated using data of species and presented in Table 1. Results for different indices have been presented in Table 2.

**3.2.2 Simpson's diversity index.** Simpson's diversity index has also shown higher values for $T_2$ and$T_7$. Lower values were recorded for $T_3$ and $T_5$.Thus, Simpson's diversity index values representboth species richness and species evenness for different transect as evident from the values presented in Tables 1 and 2. Thus, it can be concluded that Simpson's diversity index havehigher values for more number of species and evenness.

**3.2.3 Shannon-Weiner index.** The overall values for Shannon-Weiner index ranged between 0.74 and 2.06. transects $T_3$ and $T_2$have higher Shannon-Weiner index values whereas lower values have been recorded for transect $T_3$ and $T_5$.The values recorded in this index characterize the relative distribution of species in different transects.

**3.2.4 Pielou's index.** Pielou's index (J) values recorded for transectsT1 to T10 ranged between 0.67–0.99.The higher eveness value of Pielou's index wererecorded at $T_1$ and $T_2$ showing higher species individuals number and even distribution of each species in the transect.

## 3.3 Pattern of species richness

The habitat of a species is evaluated by using Simpson's diversity index *(D)*. This takes into account the number of individuals of each species [56]). Species richness in Simpson's diversity index measures the population size of each of the species present. Higher is the value of species index under Simpson's diversity *(D)*, more will be the number of species individuals present in total population. The results obtained for transects$T_7$, $T_2$ and $T_6$ clearly show this relation. Fig 5 shows that the species richness is high where the number of individualtransects are more.

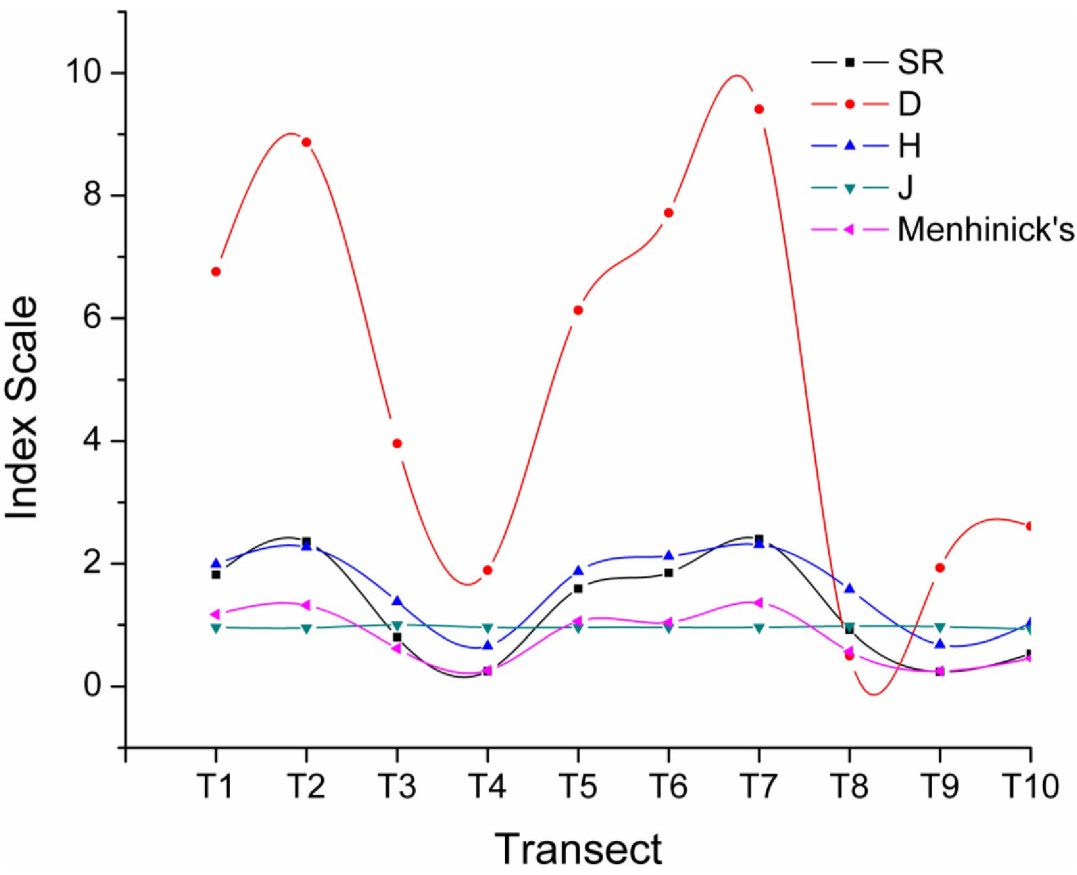

**Fig 5. Transect-wise graphical representation of diversity indices.**

Shannon Weiner index (H') assumes that heterogeneity depends upon both the number of species and their relative individual distribution. The total number of different species individuals present in an area is the measure of the total distribution of richness [57]. Pielou'sindex (J) is more suited for species closeness to environment. The presence of many individuals intransects and the maximum evenness is measured by plotting the individual species with indices [44]. To sum up, four species diversity indices were used for assessing species richness, evenness and diversity in 10 transect of MSWS. The Shannon-Weiner Index value for MSWS forest types were found to be lower in comparison to other such tropical landscapes. The methodology adopted in this study will be instructive for future researches in various forest ecosystems at spatial scales.

### 3.4 Patterns of species richness

Recorded values of species richness with different indices are tabled in Tables 1 and 2. The different indices were used to measure their values on the basis of spatial scale. To find a perfect match with the remotely sensed data and species diversity indices are difficult. The data collected on ground truth as well as google earth are indigenous to calculate the unit pixel value that the spectral heterogeneity should be the less for species diversity (Fig 6). The habitat of a species is calculated with the help of Simpson's diversity index *(D)*. This is taken account to the number of species present as the abundance of each species [56]. The species richness measured through the Simpson's diversity index is to mainly measure the population size of each of the species present. The graphical representation of the transects with respect to the diversity indices shows the species density as the number of transects, where number of individual

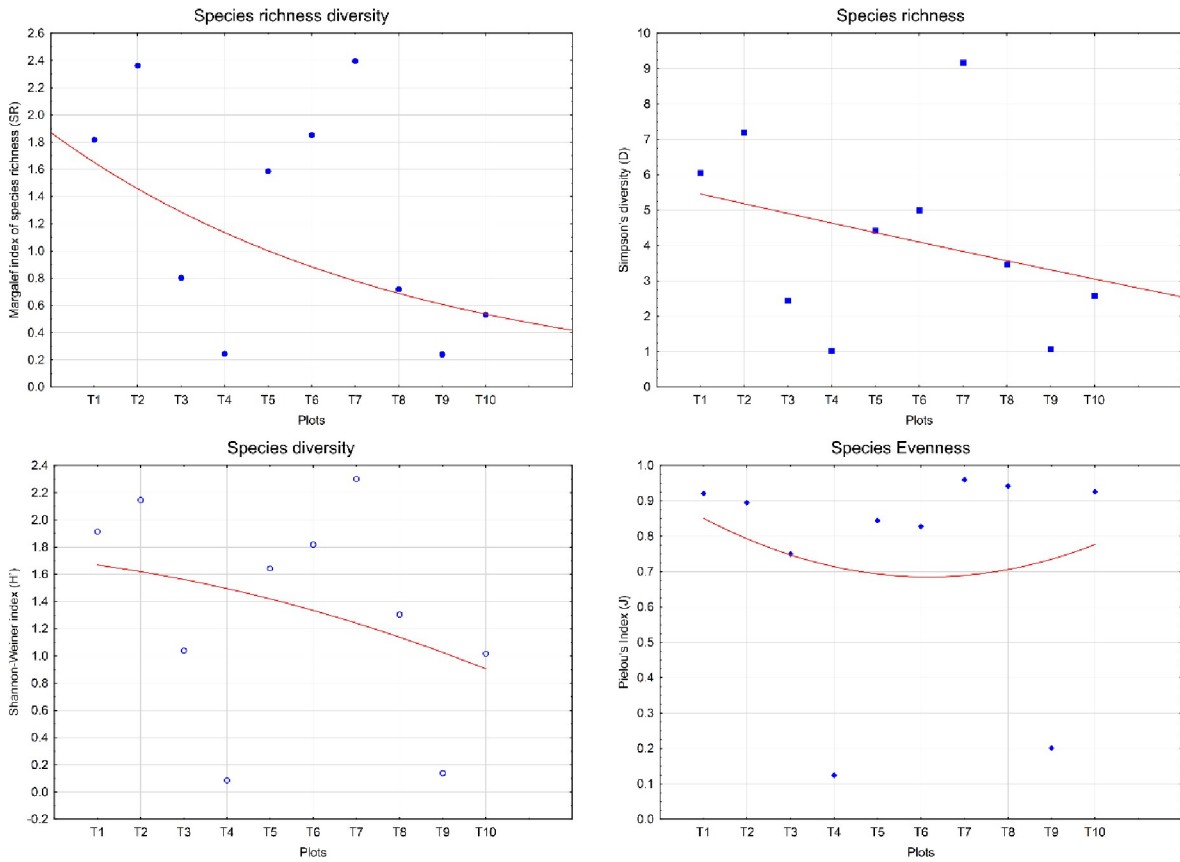

**Fig 6. Patterns graph of species richness with each transect.**

is more, the number of species richness is also high. Another indices is using here is Shannon Weiner index (*H'*), which is the diversity measure from informational theory within a particular system or transects. The total number of different species individual present in a which is the measure of total distribution of richness accounts as it is assumed that the individuals present in an area indicates the importance to diversity [57–59]. The relative importance of Shannon Weiner index *(H')* as compared to others that the individuals categorised to measure this index depends the extent of makeup of the community that our study in biofilm can be pioneer to measure the evenness of species richness. The distribution of species evenness is introduced through plotting a graph between individual per transects with respect to diversity index *(H')*.

The Margalef Species Richness index *(SR)* is the simplest measure of diversity index and this is count simply as the number of different species in any given area [44]. This data provides the demographic information about the species index in the given dataset. Margalef index categorised the abundance of diversity index plotting a graph with transects of different individual species. The presence of many individuals about any species transects and the maximum evenness is measured by plotting the individual species with indices if the Pielou's index can be measured [60].

## 3.5 Diversity dealing due to clustering

The species clustering of transect is alliedlinking with two technique. One techniques belogs to agglomerative in which the subjects have their own separate cluster. The two nearest cluster is

closest this method and species diversity is measured. The optimum cluster number of any chosen distribution is taken into account in this clustering method. Second one is the divisive methods; all the members of probable distribution are clustered and kept in one sampling. Inuniversal condition, if the p variables are measured as $X_1, X_2, X_3, X_4, \ldots, X_p$, on a species of n transects, the identified data can for topic $x$ be denoted by $x_{i1}, x_{i2}, x_{i3}, x_{i4}, \ldots, x_{jp}$ and the identified data for topic $j$ by $x_{j1}, x_{j2}, x_{j3}, x_{j4} \ldots, x_{jp}$.

The Euclidean distance between these two transects is given by Eq (10)

$$d_{ij} = \sqrt{\left(x_{i1} - x_{j1}\right)^2 + \left(x_{i2} - x_{j12}\right)^2 + \ldots\ldots\ldots + \left(x_{1p} - x_{jp}\right)^2} \qquad (10)$$

The richness values of 10 transect tree species from the clustering will be maximum and minimum and may also be the value of mean species richness. These all are calculated for the result through clustering of species. The graphical representation of the maximum, minimum and the mean species richness data are produced. The highest species richness plotted with assigned to the 10 transects as calculated with the clusters (k), but this value is increasing with respect to the k-value. The mean species richness values are also increased with increase in transects.

The clusters divided into difference species richness are approximately in between a specific value. Therefore the range is selected within the clustering. The Shannon-Weiner Diversity Index (H′) is calculated from the clustering showed a very good agreement the value derived from field species. The transects of SR and D is of the very low values with the different plants. The increase of clusters (k), do not seem to have affect the result with the explanation of the smallest k-value. The clustering results of the H′ is increase with k. The data field result reaches similar values with the index clustering value for the field data when k reaches value of 9.5. The assigned value of lower are the plantation transects for the field data, the larger the indigenous values the larger the index values (Fig 7).

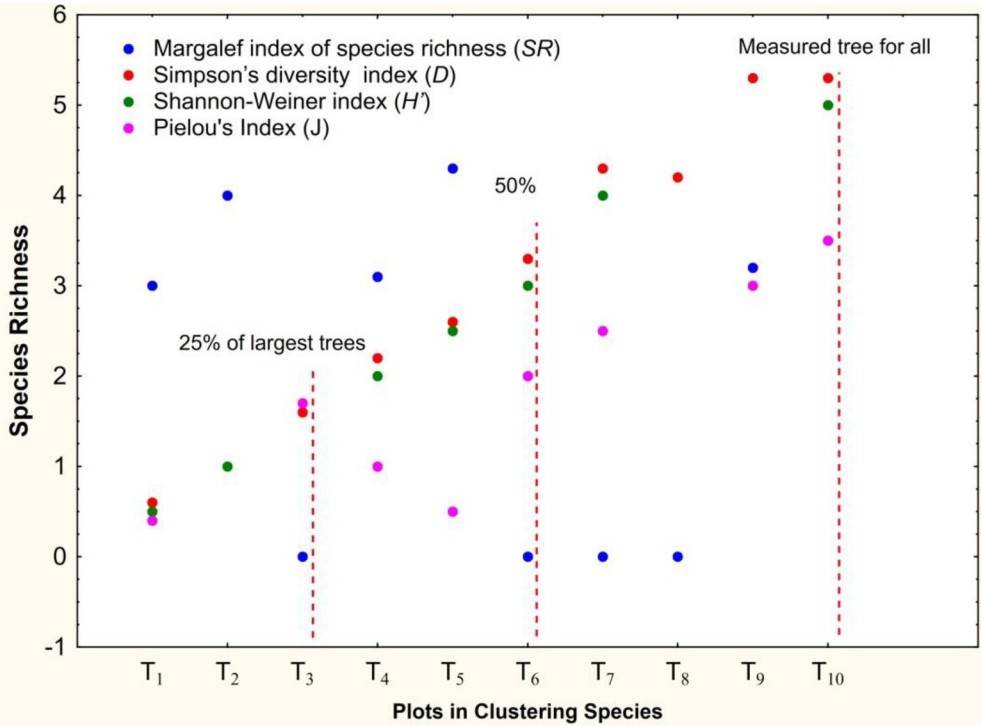

**Fig 7. Species richness for clustering with k values.**

## 4. Conclusion

Findings indicated that mixed species diversity than monoculture plantation contributes more diversity in MSWS. There is a huge demand for the broad range information and status on the forest, which can be useful for monitoring the progress towards the sustainable forest management. The present study makes an attempt to identify the concentration of species among tree diversity in MSWS.Fourimportant ecological indicator indices namely Shannon-Weiner index ($H'$), Simpson's diversity ($D$), Margalef index ($SR$) andPielou's (J) indices were make the most for species diversitymeasurment. Finding revealed that D indices was more appropatefor determining species diversity while H$'$ indices was found to be more suited for assessing species richness.

## Supporting information

**S1 Data.**
(XLSX)

## Author Contributions

**Conceptualization:** Pavan Kumar, Amey Kale.

**Data curation:** Manmohan Dobriyal, Amey Kale.

**Formal analysis:** Manmohan Dobriyal.

**Investigation:** Pavan Kumar.

**Methodology:** Pavan Kumar.

**Project administration:** Pavan Kumar.

**Resources:** Pavan Kumar.

**Software:** Pavan Kumar, Elizabeth Thounaojam.

**Supervision:** A. K. Pandey, Elizabeth Thounaojam.

**Validation:** Pavan Kumar, A. K. Pandey.

**Visualization:** A. K. Pandey, R. S. Tomar.

**Writing – original draft:** Pavan Kumar, R. S. Tomar.

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
