## [Decision Letter · Decision Letter 0]

5 Jan 2022

PONE-D-21-31788Calculating Forest Diversity with Information-Theory based Indices of Mahavir Swami Wildlife SanctuaryPLOS ONE

Dear Dr. Pavan Kumar,

Thank you for submitting your manuscript to PLOS ONE. After careful consideration, we feel that it has merit but does not fully meet PLOS ONE’s publication criteria as it currently stands. Therefore, we invite you to submit a revised version of the manuscript that addresses the points raised during the review process.

ACADEMIC EDITOR: Both the reviewers suggested for minor revision. Kindly incorporate the suggestions.

We look forward to receiving your revised manuscript.

Kind regards,

Randeep Singh

Academic Editor

PLOS ONE

Journal Requirements:

Reviewers' comments:

Reviewer's Responses to Questions

**Comments to the Author**

1. Is the manuscript technically sound, and do the data support the conclusions?

Reviewer #1: Yes

Reviewer #2: Yes

2. Has the statistical analysis been performed appropriately and rigorously? 

Reviewer #1: Yes

Reviewer #2: Yes

3. Have the authors made all data underlying the findings in their manuscript fully available?

Reviewer #1: Yes

Reviewer #2: Yes

4. Is the manuscript presented in an intelligible fashion and written in standard English?

Reviewer #1: Yes

Reviewer #2: Yes

5. Review Comments to the Author

Reviewer #1: The article spotlights on Calculating Forest Diversity with Information-Theory based Indices of Mahavir Swami Wildlife Sanctuary. The entire study is meaning and well presented. Suitable for publication after minor revision.

The title should be comprehensive and cover all steps of investigation.

Abstract should be facile and understandable to readers with technical words.

Most recent work should be cited in introduction.

Line 150-151: show the research gap of Calculating Forest Diversity in study area that leads the author to conduct the present research.

Material section similar contents if merged together

Blurry figures so, resolve this problem and also revise the figure captions with detailed illustrations

Forest Diversity is key indicator of study should be properly defined in results.

Reviewer #2: This study aims to focus on using four species diversity indices for assessing species richness, evenness and diversity in 10 transects of Mahavir Swami Wildlife Sanctuary. However, after carefully examining your manuscript, I am pleasure to inform you that it can be further considered after minor revision for publication. You may also consider the following:

1. The main method used in this study are data collection and indices calculation.

2. What are the innovations? I cannot see.

3. A comprehensive literature is required for identifying the research gaps and highlight the necessity for carrying out this study.

6. PLOS authors have the option to publish the peer review history of their article (what does this mean?). If published, this will include your full peer review and any attached files.

Reviewer #1: **Yes: **Shoaib Ahmad Anees, Ph.D

---

## [Author Response · Author response to Decision Letter 0]

30 Mar 2022

Reviewer 1

Comments

The title should be comprehensive and cover all steps of investigation.

Justification: Title has been revised according to cover all steps of investigation.

Calculating forest species diversity with information-theory based indices using sentinel-2A sensor’s of mahavir swami wildlife sanctuary.

Comments

Abstract should be facile and understandable to readers with technical words.

Justification: Abstract has been revised according to understandable to readers with technical words.

Tropical forest serves as an important pivotal role in terrestrial biological diversity. The present study makes an attempt to identify the concentration of species among tree diversity in MSWS. Four important ecological indicator indices namely Shannon-Weiner index (H’), Simpson’s diversity (D), Margalef index (SR) and Pielou's (J) indices were make the most for species diversity measurement. The research outcomes revealed that Shannon-Weiner diversity index (H/) was found to be the best index for assessing species richness while Simpson's diversity (D) index was more suited for determining species diversity. The Shannon-Weiner index value calculated for different transects not only represent the species richness but also the species evenness in each transect. The potential application of forest diversity can be used a mechanism for forest management. The methodology will retrofit better policy implementation for maintaining the health of forest species in mahavir swami wildlife sanctuary and can be applied on other reserve forest of socio-ecological significance.

Comments

Most recent work should be cited in introduction.

Justification: Recent RoL has been cited in introduction part.

a. Wang, R.; Gamon, J.A. Remote sensing of terrestrial plant biodiversity. Remote Sens. Environ. 2019, 231, 111218.

b. Turner, W.; Rondinini, C.; Pettorelli, N.; Mora, B.; Leidner, A.K.; Szantoi, Z.; Buchanan, G.; Dech, S.; Dwyer, J.; Herold, M.; et al. Free and open-access satellite data are key to biodiversity conservation. Biol. Conserv. 2015, 182, 173–176

c. Alleaume, S.; Dusseux, P.; Thierion, V.; Commagnac, L.; Laventure, S.; Lang, M.; Féret, J.-B.; Hubert-Moy, L.; Luque, S. A generic remote sensing approach to derive operational essential biodiversity variables (EBVs) for conservation planning. Methods Ecol. Evol. 2018, 9, 1822–1836.

d. Rocchini, D.; Boyd, D.S.; Féret, J.-B.; Foody, G.M.; He, K.S.; Lausch, A.; Nagendra, H.; Wegmann, M.; Pettorelli, N. Satellite remote sensing to monitor species diversity: Potential and pitfalls. Remote Sens. Ecol. Conserv. 2016, 2, 25–36.

Comments

Line 150-151: show the research gap of Calculating Forest Diversity in study area that leads the author to conduct the present research.

Justification: We completely agree and have updated the sentence accordingly.

The present study makes an attempt to calculate species diversity indices and structural forms of the tropical forest. 

Comments

Material section similar contents if merged together

Justification: We completely agree and have updated the section accordingly 

Section 2.2 and 2.3 has been merged. 

Sentinel-2A data based biodiversity extent

Comments

Blurry figures so, resolve this problem and also revise the figure captions with detailed illustrations

Justification: We completely agreed all figures have been imported up to 300 DPI. May be in PDF version it show some low resolution. 

Comments

Forest Diversity is key indicator of study should be properly defined in results.

Justification: The manuscript was substantially revised throughout in result part.

Tropical forest acts as munificence for the life forms living in the tropics, by providing habitat conditions and natural resources. Analytical scrutiny of the forest stand is provided by the structural diversity of forest and can be sub divided into three categories; tree species diversity, tree dimension diversity and tree position diversity. Forest is the key indicator of this research.

Reviewer: 2

Comments

The main method used in this study are data collection and indices calculation.

Justification: We completely agree and have updated the methodology.

Comments

What are the innovations? I cannot see.

Justification: I have removed this world from the manuscript.

Comments

A comprehensive literature is required for identifying the research gaps and highlight the necessity for carrying out this study.

Justification: Recent RoL has been cited in introduction part.

e. Wang, R.; Gamon, J.A. Remote sensing of terrestrial plant biodiversity. Remote Sens. Environ. 2019, 231, 111218.

f. Turner, W.; Rondinini, C.; Pettorelli, N.; Mora, B.; Leidner, A.K.; Szantoi, Z.; Buchanan, G.; Dech, S.; Dwyer, J.; Herold, M.; et al. Free and open-access satellite data are key to biodiversity conservation. Biol. Conserv. 2015, 182, 173–176

g. Alleaume, S.; Dusseux, P.; Thierion, V.; Commagnac, L.; Laventure, S.; Lang, M.; Féret, J.-B.; Hubert-Moy, L.; Luque, S. A generic remote sensing approach to derive operational essential biodiversity variables (EBVs) for conservation planning. Methods Ecol. Evol. 2018, 9, 1822–1836.

h. Rocchini, D.; Boyd, D.S.; Féret, J.-B.; Foody, G.M.; He, K.S.; Lausch, A.; Nagendra, H.; Wegmann, M.; Pettorelli, N. Satellite remote sensing to monitor species diversity: Potential and pitfalls. Remote Sens. Ecol. Conserv. 2016, 2, 25–36

---

## [Editor Report · Decision Letter 1]

21 Apr 2022

Calculating forest species diversity with information-theory based indices using sentinel-2A sensor’s of mahavir swami wildlife sanctuary

PONE-D-21-31788R1

Dear Dr. Pawan,

We’re pleased to inform you that your manuscript has been judged scientifically suitable for publication and will be formally accepted for publication once it meets all outstanding technical requirements.

Kind regards,

Randeep Singh

Academic Editor

PLOS ONE
---

## [Editor Report · Acceptance letter]

25 Apr 2022

PONE-D-21-31788R1 

Calculating forest species diversity with information-theory based indices using sentinel-2A sensor’s of Mahavir Swami Wildlife Sanctuary 

Dear Dr. Kumar:

I'm pleased to inform you that your manuscript has been deemed suitable for publication in PLOS ONE. Congratulations! Your manuscript is now with our production department. 

Kind regards, 

on behalf of

Dr. Randeep Singh 

Academic Editor

PLOS ONE